# The miR-141/200c-STAT4 Axis Contributes to Leukemogenesis by Enhancing Cell Proliferation in T-PLL

**DOI:** 10.3390/cancers15092527

**Published:** 2023-04-28

**Authors:** Moritz Otte, Johanna Stachelscheid, Markus Glaß, Linus Wahnschaffe, Qu Jiang, Waseem Lone, Aleksandr Ianevski, Tero Aittokallio, Javeed Iqbal, Michael Hallek, Stefan Hüttelmaier, Alexandra Schrader, Till Braun, Marco Herling

**Affiliations:** 1Department I of Internal Medicine, Center for Integrated Oncology, Aachen-Bonn-Cologne-Duesseldorf, University of Cologne, 50937 Cologne, Germany; moritz.otte1@uk-koeln.de (M.O.); johanna.stachelscheid@uk-koeln.de (J.S.); linuspaul.wahnschaffe@ukmuenster.de (L.W.); michael.hallek@uni-koeln.de (M.H.); alexandra.schrader@inserm.fr (A.S.); till.braun@uk-koeln.de (T.B.); 2Section for Molecular Cell Biology, Institute of Molecular Medicine, Faculty of Medicine, Martin Luther University Halle-Wittenberg, Charles Tanford Protein Center, 06120 Halle, Germany; markus.glass@medizin.uni-halle.de (M.G.);; 3Department of Hematology, Cellular Therapy, and Hemostaseology, University of Leipzig, 04103 Leipzig, Germany; qu.jiang@medizin.uni-leipzig.de; 4Department of Pathology and Microbiology, University of Nebraska Medical Center, Omaha, NE 68198, USA; waseem.lone@unmc.edu (W.L.); jiqbal@unmc.edu (J.I.); 5Institute for Molecular Medicine Finland (FIMM), HiLIFE, University of Helsinki, 00014 Helsinki, Finland; aleksandr.ianevski@helsinki.fi (A.I.); tero.aittokallio@helsinki.fi (T.A.); 6Institute for Cancer Research, Oslo University Hospital, Oslo Centre for Biostatistics and Epidemiology (OCBE), University of Oslo, 0372 Oslo, Norway; 7Excellence Cluster for Cellular Stress Response and Aging-Associated Diseases, Center for Molecular Medicine Cologne, University of Cologne, 50937 Cologne, Germany; 8CIRI, Centre International de Recherche en Infectiologie, Team Lymphoma ImmunoBiology, INSERM, U1111 CNRS UMR 5308, University of Lyon, Université Claude Bernard Lyon 1, 69364 Lyon, France

**Keywords:** T-PLL, miR-141, miR-200c, STAT4, T-cell lymphoma, cell proliferation, leukemogenesis

## Abstract

**Simple Summary:**

T-prolymphocytic leukemia (T-PLL) is a rare and poor-prognostic mature T-cell leukemia. A better understanding of this chemotherapy-refractory disease is highly warranted to identify novel treatment strategies. We previously identified an elevated expression of the miR-141/200c cluster in T-PLL cells. Here, we show a pro-proliferative effect of miR-141/200c in mature T-cell lymphoma cell lines. We further characterize a miR-141/200c-driven transcriptome, entailing altered expression of genes involved in pathways regulating cell survival and differentiation. Among those, we identified *STAT4* as a miR-141/200c target gene, with low *STAT4* expression being associated with an immature phenotype of T-PLL cells and with shortened overall survival of T-PLL patients. Overall, we present an oncogenic miR-141/200c-STAT4 signaling route in T-PLL, demonstrating, for the first time, a role of non-protein-coding genes in the leukemogenesis of this devastating disease.

**Abstract:**

T-prolymphocytic leukemia (T-PLL) is a rare and mature T-cell malignancy with characteristic chemotherapy-refractory behavior and a poor prognosis. Molecular concepts of disease development have been restricted to protein-coding genes. Recent global microRNA (miR) expression profiles revealed miR-141-3p and miR-200c-3p (miR-141/200c) as two of the highest differentially expressed miRs in T-PLL cells versus healthy donor-derived T cells. Furthermore, miR-141/200c expression separates T-PLL cases into two subgroups with high and low expression, respectively. Evaluating the potential pro-oncogenic function of miR-141/200c deregulation, we discovered accelerated proliferation and reduced stress-induced cell death induction upon stable miR-141/200c overexpression in mature T-cell leukemia/lymphoma lines. We further characterized a miR-141/200c-specific transcriptome involving the altered expression of genes associated with enhanced cell cycle transition, impaired DNA damage responses, and augmented survival signaling pathways. Among those genes, we identified *STAT4* as a potential miR-141/200c target. Low *STAT4* expression (in the absence of miR-141/200c upregulation) was associated with an immature phenotype of primary T-PLL cells as well as with a shortened overall survival of T-PLL patients. Overall, we demonstrate an aberrant miR-141/200c-STAT4 axis, showing for the first time the potential pathogenetic implications of a miR cluster, as well as of STAT4, in the leukemogenesis of this orphan disease.

## 1. Introduction

T-prolymphocytic leukemia (T-PLL) is a rare (0.6/1 million/year), but aggressive, T-cell malignancy [1,2]. T-PLL commonly presents with high blood lymphocyte counts (>100G/L), bone marrow infiltration, and splenomegaly [3]. Current treatment options for T-PLL are limited, as the responses to conventional chemotherapies are poor. First-line therapy in most cases consists of the monoclonal anti-CD52 antibody Alemtuzumab, with response rates to single-agent inductions of >80% [4]. However, virtually all patients relapse thereafter if not consolidated. Long-term remissions can be achieved in a fraction of patients, who received an allogeneic stem cell transplantation, but with only a subset of patients (30–50%) being eligible for this procedure. Over all the patient subsets, the median overall survival (OS) of T-PLL is less than two years [1].

T-PLL cells carry complex karyotypes with rearrangements of chromosome 14q, leading to constitutive expression of the proto-oncogene T-cell leukemia 1A (*TCL1A*) as the most common lesion [5]. Other recurrent genomic alterations include ataxia telangiectasia mutated (*ATM*) deletions/mutations as well as *MYC* amplifications [6]. In addition, we recently identified Janus kinase (*JAK*)/signal transducer and activator of transcription (*STAT*) genomic alterations as a hallmark of T-PLL’s genomic landscape [7]. Gain-of-function mutations as well as genomic losses of negative regulators of JAK/STAT signaling, as seen in 90% [7] of T-PLL cases, lead to constitutive activation of the JAK1/JAK3/STAT5B axis. The involvement of STAT4, which is an important mediator of T-cell differentiation [8], has not been shown in the pathogenesis of T-PLL before.

Frequent gains on chromosome 8q involve, in addition to *MYC,* the argonaute RISC catalytic component 2 (*AGO2*) [6]. AGO2 is the main regulator of microRNA (miR) activity, possessing endonuclease activity and acting as the executor of the RNA-induced silencing complex (RISC) [9]. Besides AGO2′s role in mediating post-transcriptional mRNA degradation, we recently identified an AGO2-arbitrated activation of oncogenic T-PLL cells through signal-amplifying protein–protein interactions [10].

In the past years, miRs have emerged as important regulators of cell fate. MiRs are small non-coding RNAs with an average length of 22 nucleotides [11]. By binding to the 3′ untranslated region (UTR) of target mRNAs, miRs function as post-transcriptional repressors within the RISC [12]. In cancer, miRs can exert both oncogenic (‘onco-miR’) or tumor-suppressive functions and have been causally implicated in mature B- and T-lymphoid malignancies. As a prominent example, downregulation of the miR-15/16 cluster in chronic lymphocytic leukemia (CLL) has been shown to be critical for B-cell leukemogenesis [13,14]. Moreover, some miRs have been identified to play a role in the pathogenesis of mature T-cell neoplasms, such as miR-29 and -200, in cutaneous T-cell lymphoma (CTCL) [15,16] and miR-150 in NK/T-cell lymphoma [17]. Thus, dissecting miR deregulations and their functional effects in carcinogenesis is of high interest to both researchers and clinicians. Several clinical trials using miR replacement therapy in cancer have been initiated, although none have been approved for clinical practice so far [18].

In a combined approach of small-RNA and transcriptome sequencing of 41 T-PLL patients, we recently characterized the miR-ome of T-PLL cells [19]. We detected a T-PLL-specific miR expression signature, consisting of 34 deregulated miRs when compared to healthy donor-derived T cells. As a hallmark, miRs of the miR-141/200c cluster were among the highest abundant miRs in T-PLL, separating the cases into two subgroups, one with high and one with low miR-141/200c expression [19]. The miR-141/200c cluster belongs to the miR-200c family, consisting of five different members (miR-141, miR-200a/b/c, and miR-429) [20]. The miR-200 family is mainly implicated in epithelial-to-mesenchymal transition by regulating the expression of Zinc Finger E-box-binding homeobox 1/2 (*ZEB1/2*) [21]. Deregulations of miR-141/200c have raised the interest of researchers within the past decade, as cell-type-dependent roles like onco-miRs and tumor-suppressive miRs have been claimed [22]. In T-PLL, upregulation of miR-141/200c was shown to be associated with aberrant transforming growth factor beta (TGFβ) signaling [23].

Here, we discovered an association of accelerated proliferation and reduced stress-initiated cell death induction with elevated miR-141/200c levels in T-PLL by utilizing stable miR-141/200c-overexpressing T-cell leukemia/lymphoma lines. We further characterized a miR-141/200c-specific transcriptome in T-PLL cells as well as in miR-141/200c-overexpressing HuT78 cells. Finally, we identified *STAT4* as a potential novel target of miR-141-3p, with low *STAT4* expression being associated with poor survival of T-PLL patients. This, for the first time, emphasizes a potential role of STAT4 for T-PLL’s leukemogenesis, besides the commonly affected pro-oncogenic JAK1/JAK3/STAT5B axis [7].

## 2. Materials and Methods

### 2.1. Patient Cohort

Primary cell isolates of 103 T-PLL patients (detailed patient characteristics are given in Appendix A) and of T cells from 6 age-matched healthy donors were studied for clinical characteristics (*n* = 103) and subjected to poly-A-RNA (*n* = 50) and small RNA sequencing (*n* = 46) [19]. The diagnosis of T-PLL was confirmed according to WHO criteria and consensus guidelines [3,24]. All patients provided informed consent according to the Declaration of Helsinki. The collection and use of samples were approved for research purposes by the Ethics Committee of the University Hospital of Cologne (#11–319). Data on *STAT4* mRNA expression in angioimmunoblastic T-cell lymphoma (AITL, *n* = 104), anaplastic large T-cell lymphoma (ALCL; *n* = 40 ALK^−^; *n* = 29 ALK^+^), and peripheral T-cell lymphoma (PTCL; *n* = 35 PTCL-GATA3; *n* = 38 PTCL-TBX21), compared to pooled CD4^+^ and CD8^+^ T cells, were extracted from published transcriptome sequencing datasets [25,26].

### 2.2. Cell Culture

Suspension cultures of the cell lines HuT78, HH, MOLT-4, and SUP-T11, as well as primary tumor cell isolates of T-PLL patients and healthy donor-derived T cells, were maintained according to standard protocols (for detailed information on the origin of the used cell lines and cell culture conditions, see Appendix A). For serum starvation experiments, the concentration of fetal bovine serum (FBS) was reduced to one-tenth (1% for HuT78 and MOLT-4; 2% for HH and SUP-T11). All cell lines were frequently tested for *Mycoplasma* infection by standard PCR protocols. For details on the assessment of cell density and viability, as well as mixed culture and cell proliferation experiments, see Appendix A.

### 2.3. Stable miR-141/200c-Overexpressing Cell Lines

To induce stable miR-141/200c overexpression in T-cell leukemia/lymphoma lines, commercially available pLenti4.1ExmiR-200c-141 (third-generation lentiviral vector expressing the miR-141/200c genomic cluster) and pLenti4.1ExmiR-200b-200a-429 (used as a backbone for the empty control vector) were purchased [21]. Details on plasmid mutagenesis and on lentiviral transduction into the cell lines HuT78, HH, MOLT-4, and SUP-T11, which were performed in line with standard protocols, are given in the Appendix A.

### 2.4. RNA Isolation and Sequencing, Data Processing and Analyses, and miR Target Prediction

Details on RNA isolation, RNA sequencing, data processing, and data analyses of primary T-PLL cells are published [19]. Total RNA from the cell lines HuT78^empty^ and HuT78^miR-141/200c^ was isolated using the miRVana kit (ThermoFisher, Waltham, MA, USA; for qRT-PCR of miR-141/200c) and the RNeasy Plus kit (Qiagen, Venlo, Netherlands, for RNA sequencing) according to the manufacturer’s instructions. RNA quality and concentration were assessed using the 4150 TapeStation (Agilent, Santa Clara, CA, USA), and samples with an RNA integrity number <6 were excluded. Early passages of the cell lines HuT78^empty^ and HuT78^miR-141/200c^ were kept in normal culturing conditions for one week and were then subjected to serum starvation. Three replicates of both cell lines, each at day 0 and after 3 days of serum starvation, were subjected to library preparation and sequenced on the Illumina TruSeq platform (Illumina, San Diego, CA, USA) according to the manufacturer’s instructions for polyA-RNA sequencing. Gene set enrichment analysis (GSEA) was performed using the R-package clusterProfiler (v 4.1.4 [27]) and MSigDB gene sets (v7.4 [28]) utilizing the fgsea algorithm and setting the exponent parameter to 0 for unweighted analyses of log2 fold change (fc) sorted gene lists obtained from differential gene expression analyses. The prediction strategy for putative mRNA targets for the miR-141/200c cluster samples has been described previously [19]. We included genes with a negative correlation in primary T-PLL cells (rho < 0, *p* < 0.05) as well as in miR-141/200c-overexpressing HuT78 cells (at day 0 and day 3; fc < 0.5, FDR < 0.01)

### 2.5. Western Blot and Quantitative Real-Time PCR

Western blots on whole-cell protein lysates and quantitative real-time PCR on total RNA were performed according to standard protocols and are described in detail in the Appendix A.

### 2.6. Assessment of Associations with Surface Marker Expression and Clinical Data

Association of *STAT4* mRNA expression with surface marker expression (assessed by flow cytometry) was performed in a cohort of *n* = 25 T-PLL cases (divided by median *STAT4* expression), which was previously characterized by transcriptome sequencing [19]. To assess the impact of *STAT4* expression on overall patient survival, we utilized a previously published, gene expression array-characterized cohort of *n* = 63 T-PLL cases [6]. A log-rank test of the overall survival probability of T-PLL patients (divided by median *STAT4* expression) and a cox regression-based multivariate survival analysis, including *STAT4* expression, the white blood cell count, and the Cumulative Illness Rating Scale (CIRS) at diagnosis, were performed.

## 3. Results

### 3.1. The Micro-RNA Cluster miR-141/200c Is Highly Upregulated in T-PLL Cells and Is Associated with a Gene Signature of Cell Cycle Acceleration and Enhanced Survival Signaling

Initially, we sought to characterize the elevated expression of the miR-141/200c cluster as well as its implications on transcriptome signatures in T-PLL. To this end, we recapitulated our previously collected small RNA sequencing data of isolated peripheral blood mononuclear cells (PBMCs; median: 95.4% leukemic cell purity) of 46 T-PLL patients and of CD3^+^ pan-T cells from 6 age-matched healthy donors [19]. MiR-141-3p (fc = 43.2, FDR = 0.005) and miR-200c-3p (fc = 38.2, FDR = 0.005) stood out as two of the most highly upregulated miRs (third- and fourth-highest fcs of all significantly deregulated miRs, respectively) in T-PLL relative to healthy donor-derived T cells. In addition, miR-141-3p and miR-200c-3p showed high overall abundance, with miR-141-3p having the highest absolute counts per million (CPM) among all deregulated miRs in T-PLL (mean CPM across all T-PLL; Figure 1a). Notably, unsupervised hierarchical clustering revealed two subgroups of T-PLL cases, which were mainly divided by their miR-141/200c expression levels. While half of all sequenced cases displayed miR-141/200c expression similar to healthy donor-derived T cells (‘low miR-141/200c’ cluster, *n* = 23 cases), the other half showed marked upregulation of both miR-141-3p and miR-200c-3p (‘high miR-141/200c’ cluster, *n* = 23 cases; Figure 1b). 

Having identified this dichotomy in miR-141/200c expression in T-PLL, we associated the expression of the miR-141/200c cluster with mRNA sequencing data in a cohort of 41 T-PLL cases, which were subjected to both small RNA and mRNA sequencing. Unsupervised hierarchical clustering revealed a distinct transcriptome in the ‘high miR-141/200c’ cohort, consisting of 356 differentially expressed genes discriminating between the ‘high’ and ‘low’ miR-141/200c cluster (Figure 1c, Appendix A). ‘High miR-141/200c’-expressing T-PLL cells showed an elevated expression of genes involved in oncogenesis, such as the immune-modulatory receptor tyrosin kinase *MST1R* (fc = 8.71, FDR < 0.0001) [29] and the chemokine *CCL25* (fc = 59.77, FDR < 0.0001) [30], as well as genes contributing to cell cycle progression such as *CCNA1* (fc = 51.14, FDR < 0.0001) [31]. As miR-mediated regulation of cellular processes is based on the induction of low-level changes of mRNA abundances affecting a larger cohort of genes rather than one single factor, we conducted gene set enrichment (GSEA) analyses using the complete list of fcs of all protein-coding genes characterizing the ‘high miR-141/200c’ cohort (Figure 1d). In total, 14 of the 50 MsigDB hallmark gene sets displayed significant positive enrichment (FDR < 0.05) in high miR-141/200c-expressing T-PLL as compared to miR-141/200c-low T-PLL, whereas 12 of 50 gene sets showed significant negative enrichment in the high miR-141/200c patient subset. Exemplarily, the *HALLMARK* pathways *E2F_TARGETS*, *DNA_REPAIR,* and *OXIDATIVE_PHOSPHORYLATION* stood out as significantly altered in association with miR-141/200c expression (Figure 1e). In line with the role of miR-141/200c in regulating *ZEB1/2* expression, the gene set *EPITHELIAL_MESENCHYMAL_TRANSITION* showed negative enrichment in the ‘high miR-141/200c’ cohort.

### 3.2. Elevated Expression of miR-141/200c Leads to Enhanced Cell Proliferation and Reduced Stress-Induced Cell Death under Serum Starvation in Mature T-Cell Leukemia/Lymphoma Lines

To identify a cellular phenotype associated with miR-141/200c upregulation, we created miR-141/200c-overexpressing T-cell leukemia/lymphoma lines. In the absence of available genuinely T-PLL-derived human cell lines, we used here the lines HuT78 and HH (mature; from CTCL) as well as MOLT-4 and SUP-T11 (naïve; from T-acute lymphoblastic leukemia (T-ALL)) for further studies. As basal expression of the miR-141/200c cluster was (in contrast to T-PLL) generally low in the T-cell leukemia/lymphoma cell lines and in healthy donor T cells, we opted for overexpression of miR-141/200c (instead of knocking it down) in the T-cell lines to induce a T-PLL-like miR-141/200c-related gene expression signature (Appendix A). For this, we introduced a vector leading to stable miR-141/200c overexpression or an empty control vector into these cell lines via lentiviral transduction (see Appendix A for details on plasmid mutagenesis). Notably, miR-141/200c-overexpressing HuT78 and HH cells exhibited an enhanced cell proliferation (HuT78: *p* < 0.0001; HH: *p* = 0.02; two-way ANOVA) accompanied by higher viability in serum-starved conditions (HuT78: *p* < 0.0001; HH: *p* = 0.01; two-way ANOVA), compared to their respective empty-vector controls (Figure 2a,b). To further substantiate the pro-proliferative effects of miR-141/200c overexpression, we performed competitive mixed cultures of miR-141/200c-transduced or empty-vector HuT78 cells (and, therefore, GFP-positive cells) with wild-type (wt, GFP-negative) HuT78 cells in serum-starved culture medium.

Again, miR-141/200c-overexpressing cells outgrew the wt control cells, while the empty-vector carrying cells did not show that effect (Figure 2c). Concurrently, we identified an accelerated reduction of the intensity of the Cell Proliferation Dye eFluor^TM^ 670 in miR-141/200c-overexpressing HuT78 cells, compared to their empty-vector control cells, corresponding to an increased proliferation rate as one of the reasons for the aforementioned differences in cell densities (Figure 2d). In contrast to the effects identified in the mature T-cell lymphoma lines, we did not observe any impacts of overexpressed miR-141/200c on proliferation and viability in the immature T-ALL cell lines MOLT-4 and SUP-T11 (Appendix A), indicating that the identified effects potentially depend on the T-cell differentiation state of the targeted cell line.

### 3.3. Overexpression of miR-141/200c Shapes a Pro-Oncogenic Transcriptome of the Mature T-Cell Lymphoma Line HuT78, Resembling the Alterations Seen in T-PLL

Having identified a pro-proliferative and more stress starvation-resistant cellular phenotype upon miR-141/200c overexpression, we next analyzed gene expression signatures associated with the upregulation of this miR cluster, potentially mediating the observed effects. For this purpose, we utilized our miR-141/200c-overexpressing and empty-vector control HuT78 cells and performed next-generation polyA-RNA sequencing of samples without cell culture effect (d0) and after three days in serum-starved cultures (d3, see Section 2 for details on sample acquisition). Principal component analysis (PCA) revealed miR-141/200c overexpression as well as serum starvation as major determinants of transcriptomic alterations (Figure 3a). By comparing miR-141/200c-overexpressing HuT78 to the empty-vector control HuT78 cells, we identified *n* = 530 (d0)/*n* = 597 (d3) significantly downregulated (fc < 0.5, FDR < 0.01) and *n* = 450 (d0)/*n* = 604 (d3) significantly upregulated genes (fc > 2, FDR < 0.01) in the miR-141/200c-overexpressing condition. Exemplarily, genes involved in JAK/STAT pathways, such as *SOCS3* (fc = 0.24, FDR < 0.0001) [32] and *STAT4* (fc = 0.25, FDR < 0.0001) [33], or in cell cycle regulation, such as *CCND1* (fc = 9.68, FDR < 0.0001) [31], were differentially expressed when comparing HuT78^empty^ vs. HuT78^miR-141/200c^ cells at d0 (Figure 3b and Appendix A, Appendix A). Analogous to our previous approach, we next conducted GSEA comparing HuT78^empty^ vs. HuT78^miR-141/200c^ cells at the two time points to depict global changes in cellular pathways. Comparable with the alterations seen in T-PLL samples, differentially enriched pathways were associated with cell cycle regulation, DNA repair, inflammatory responses, or survival signaling and mostly concordant between samples at d0 and d3 (Figure 3c and Appendix A). Again, the epithelial-mesenchymal transition gene set showed significant negative enrichment in the miR-141/200c-overexpressing cells, in line with the well-established miR-141/200c function in regulating *ZEB1/2*. Additionally, we investigated the effects of serum starvation on the transcriptome of both cell lines. We identified a specific gene expression signature upon starvation, affecting pathways of cell cycle progression, apoptosis induction, and cellular metabolism (Appendix A, Appendix A). Interestingly, some gene sets, such as *P53_PATHWAY* and *OXIDATIVE_PHOSPHORYLATION,* displayed significant negative enrichment at d3 compared to d0 in HuT78^miR-141/200c^, while being positively enriched in HuT78^empty^; hinting at differences in how the cellular transcriptome shifts upon serum starvation in dependence of miR-141/200c levels (Appendix A).

Next, we aimed at identifying an overlap of differentially expressed genes upon miR-141/200c upregulation in three different contexts: MiR-141/200c-overexpressing HuT78 cells at (i) d0 and (ii) d3 (compared to empty-vector HuT78 cells), as well as (iii) T-PLL with elevated miR-141/200c expression compared to miR-141/200-low T-PLL. We identified *n* = 604 deregulated genes shared between d0 and 3, while *n* = 54 genes showed an overlap between all three conditions, indicating an overarching effect of high miR-141/200c levels on gene expression in mature T-cell malignancies (Figure 3d). Exemplarily, the tumor-suppressive cell adhesion molecule 1 (*CADM1* [34]) showed a negative association with miR-141/200c expression in HuT78 cells as well as primary T-PLL cases. We next conducted a miR-141/200c target gene prediction in T-PLL as well as in miR-141/200c-overexpressing HuT78 cells (Figure 3e). In total, we identified an overlap of 30 putative miR-141/200c target genes, with the transcription factor *STAT4* emerging as one of the predicted targets, potentially mediating the cell’s biologic changes observed upon miR-141/200c upregulation.

### 3.4. Downregulation of STAT4 by the miR-141/200c Cluster Is Associated with an Immature Cell Phenotype and Poor Patient Outcomes in T-PLL

As the miR target prediction revealed *STAT4* to be a putative target of miR-141-3p and as *STAT4* was among the top deregulated genes in miR-141/200c-overexpressing HuT78 cells, we sought to further characterize *STAT4* expression in T-PLL, focusing on its relation to miR-141/200c levels. In line with being a putative miR-141-3p target mRNA, *STAT4* gene expression negatively correlated with miR-141-3p expression in T-PLL and healthy donor-derived T-cell controls (rho = −0.36, *p* = 0.02, Figure 4a). Furthermore, *STAT4* levels were generally lower in T-PLL as compared to healthy donor CD3^+^ pan-T cells (Figure 4b), with the lowest *STAT4* expression values in the miR-141/200c-high T-PLL subset (Appendix A). We next aimed to substantiate these differences in *STAT4* mRNA abundances at the level of protein expression. For this purpose, we first compared STAT4 protein expression between low and high miR-141/200c-expressing T-PLL (Figure 4c). In line with the STAT4 mRNA expression, we identified globally diminished STAT4 protein levels in T-PLL cells, compared to healthy donor-derived T cells. Strikingly, STAT4 protein expression levels were even lower in high miR-141/200c-expressing T-PLL samples, further supporting STAT4 as a target of this miR cluster. Then, we compared STAT4 protein expression between low and high miR-141/200c-expressing T-cell leukemia/lymphoma lines (Figure 4d). Similar to the primary T-PLL cells, enforced miR-141/200c overexpression in HuT78 and HH cells led to loss of STAT4 protein expression. The immature T-ALL cell lines MOLT-4 and SUP-T11, both without an enhanced growth behavior upon miR-141/200c upregulation, did not express STAT4 in either condition, supporting a potential role of STAT4 in the observed miR-141/200c effects.

Given the main function of STAT4 in T-cell differentiation [8], we explored the surface expression of T-cell differentiation markers in T-PLL cells by flow cytometry, in correlation with their *STAT4* mRNA expression, utilizing a two-step procedure. First, we assessed associations of *STAT4* mRNA expression with the expression of singular differentiation markers. Notably, T-PLL cases with low *STAT4* expression exhibited a higher percentage of cells expressing the naïve surface markers CD45RA and CXCR3, as well as a lower percentage of cells expressing the memory surface marker CD45RO, indicating a more immature T-cell differentiation state upon *STAT4* downregulation (Figure 4e). Next, we defined phenotypes of naïve/memory T-cell differentiation via multi-parameter immunophenotyping in our T-PLL cohort. Again, we aimed at identifying associations of *STAT4* mRNA expression with the distribution into the different T-cell differentiation states. In line with the findings of single marker staining, low *STAT4* T-PLL most often displayed a naïve phenotype, whereas the majority of high *STAT4* T-PLL showed a central-memory profile (Figure 4f). In addition, we found the surface receptors CXCR3 and CXCR5 as well as the transcription factor GATA3, markers for Th1/Th2 differentiation, to be significantly higher expressed in T-PLL with low *STAT4* levels (Appendix A). 

To reveal whether *STAT4* downregulation in mature T-cell neoplasms is a phenomenon specific to T-PLL, we re-analyzed published gene expression profiling (GEP) array data from different entities of mature T-cell lymphomas (TCL) compared to either pooled CD4^+^ or CD8^+^ T cells (Figure 4g) [25,26]. There was a mild *STAT4* downregulation in angioimmunoblastic T-cell lymphoma (AITL), anaplastic large-cell lymphoma (ALCL), and peripheral T-cell lymphoma, which was not otherwise specified (PTCL-NOS). 

Finally, we observed that lower *STAT4* gene expression was significantly associated with inferior overall survival in T-PLL (Figure 5a). Furthermore, overall survival upon first-line treatment with alemtuzumab-based regimens tended to be worse in T-PLL with low *STAT4* expression (Appendix A). In a multivariate model for overall survival, additionally including white blood cell count at diagnosis and Cumulative Illness Rating Scale (CIRS, comorbidity index) at diagnosis, *STAT4* expression remained the strongest predictor for lower overall survival (Figure 5b). These results further consolidate the important role of STAT4 in the leukemogenesis of T-PLL.

## 4. Discussion

Our mechanistic disease model of T-PLL is incomplete and particularly lacks an understanding on the impact of non-protein-coding genes [6]. Erkeland and colleagues [23] as well as our group [19] recently characterized the miR-ome of T-PLL patients. Based on our finding of strong upregulation of the miRs-141 and -200c, we proposed an oncogenic role of this cluster in the leukemogenesis of T-PLL. In our combined approach of small RNA sequencing and transcriptome sequencing [19], we identified a miR-141/200c-specific gene expression signature that encompasses genes involved in accelerated cell cycle transition, enhanced survival signaling, and altered DNA damage responses. In line, we observed a faster proliferating and stress starvation-resistant phenotype in miR-141/200c-overexpressing, mature T-cell leukemia/lymphoma lines. We further identified *STAT4* as a potential target gene of the miR-141/200c cluster in these cell line models as well as in primary T-PLL. Reduced *STAT4* expression was associated with an immature T-cell differentiation state and shorter overall survival of T-PLL patients. Overall, we implicate here the contribution of a miR cluster, namely miR-141/200c, and one of its relevant targets (i.e., *STAT4*), to defined cellular phenotypes as part of the pathogenesis of T-PLL.

MiR-141-3p and miR-200c-3p stood out as having high absolute expression values in primary cells of a subset of T-PLL cases. Previously, miR-141 and -200c were implicated in the carcinogenesis of various cancer entities [20] However, the role of this cluster seemed to be cell-type-dependent: While a tumor-suppressive effect was claimed for osteosarcoma and breast cancer [35,36], an oncogenic role was demonstrated in colorectal cancer [37]. Notably, a recent meta-analysis identified an association of miR-200c expression with poorer patient survival across several cancer entities [38]. The best-studied mechanism of this miR cluster is its regulation of the epithelial-mesenchymal transition (EMT), an essential component of the invasive growth program of solid tumors [21]. MiR-141 and miR-200c target the mRNAs coding for ZEB1 and ZEB2, favoring the expression of the epithelial marker E-cadherin, and, thereby, counteracting EMT. MiR-141/200c mediated transcriptional regulation is known to further affect JAK/STAT signaling, apoptotic resistance, and survival signaling pathways (e.g., AKT signaling) [39]. In T-PLL, Erkeland and colleagues demonstrated the involvement of miR-141/200c in TGFβ signaling [23]. Here, we expand the function of miR-141/200c to a potential regulation of STAT4, as well as presenting data on accelerated proliferation and diminished stress-induced cell death upon miR-141/200c upregulation in T-cell lymphoma/leukemia models. Notably, more comprehensive associations between miR-141/200c expression and clinical characteristics were reported by our group [19] and others [23]. While we identified an association between miR-141/200c upregulation and a more activated T-cell phenotype [19], Erkeland et al. demonstrated a shortened survival of T-PLL patients with high miR-141/200c expression [23].

In contrast to other STAT molecules that are expressed by a wide variety of cell types, STAT4 is mainly expressed by immune cells, in particular, dendritic cells and T helper 1 (Th1) cells [40]. As a main function, STAT4 is important for T-cell differentiation, especially Th1 cell polarization [8]. Since cell differentiation is often associated with reduced proliferative potential, the miR-141/200c-mediated downregulation of STAT4, as observed in T-PLL, might conserve rather early differentiation stages, thereby contributing to a less restricted proliferative potential of the malignant cells. Similar to the miR-141/200c cluster, there is conflicting literature on a possible oncogenic or tumor-suppressive role of STAT4 in cancer. Notably, a tumor-suppressive function of STAT4 has been postulated for hepatocellular carcinoma as well as B-cell malignancies, being essential in cell cycle regulation and apoptosis induction [41,42,43]. In addition, downregulation of STAT4 has been reported for cutaneous T-cell lymphoma (CTCL) and was associated with progressed disease stages [44,45]. An immediate regulation of STAT4 by miR-141-3p via STAT4’s 3′UTR has been demonstrated by luciferase reporter assays in various cellular systems (e.g., hepatocellular carcinoma) [46,47]. From this, we concluded the same direct regulatory relationships to underlie our observations in mature T-cell leukemia cells without explicitly validating these molecular interactions. In T-PLL, research on aberrant JAK/STAT signaling was mainly concentrated around an activated JAK1/JAK3/STAT5 axis [7]. Here, we demonstrate data that for the first time implicate an anti-proliferative and potentially tumor-suppressive role of STAT4 in T-PLL. The association of *STAT4* downregulation with shortened patient survival is particularly intriguing as this further emphasizes its relevance in the course towards and/or of overt T-PLL.

Explaining the phenotype of miR-141/200c-overexpressing cell lines by downregulation of STAT4 requires a more detailed analysis: To validate a direct involvement of STAT4 on the miR-141/200c-mediated effect on accelerated proliferation and less stress-induced cell death, our data should be complemented by cellular systems harboring a STAT4 downregulation, which is a task for future research. However, showing the strong, negative association between the expression of the miR-141/200c cluster and STAT4 mRNA and protein, as well as given the association of STAT4 downregulation with an immature phenotype, the faster proliferating effects of miR-141/200c are likely explained by this regulatory miR-141/200c-STAT4 axis. Further studies also need to address (i) additional modes of STAT4 downregulation besides the miR-141/200c upregulation (e.g., epigenetic dysregulation), (ii) the functional impact of further miR-141/200c target genes in the pathogenesis of T-PLL (e.g., downregulation of *CADM1*, recently shown to be associated with poor prognosis in primary adult T-cell leukemia/lymphoma [34]), and (iii) STAT4 downregulation in other entities of mature T-cell lymphoma.

Besides its inherent prognostic information, our data also provide a rationale to further investigate whether the miR-141/200c-STAT4 axis can be utilized as a therapeutic target. Even though the use of miR inhibitors as therapeutic molecules is still in its infancy, a few non-human in vivo studies and clinical trials have been conducted that have shown promise, combining therapeutic efficacy with acceptable toxicity [48,49]. Thus, further investigations into the use of miR inhibitors, both as single substances and as components of drug combinations, are highly warranted.

## 5. Conclusions

Our findings unravel a pro-oncogenic role of the miR-141/200c cluster in the pathogenesis of T-PLL. In mature T-cell lymphoma lines, we identified a faster proliferating and more stress-resistant phenotype upon miR-141/200c overexpression. We further characterized a specific gene expression signature of enhanced cell cycle transition and impaired DNA damage repair in these cellular systems, and we reconciled these changes with miR-141/200c-associated alterations in primary T-PLL samples. We identified *STAT4* as a target of the miR-141/200c cluster. Finally, we demonstrated an association of miR-141/200c-driven downregulation of STAT4 with an immature phenotype as well as shortened survival in primary T-PLL cases. This further emphasizes aberrant JAK/STAT signaling as a key hallmark of T-PLL’s pathogenesis with an involvement of the identified deregulation of STAT4 besides the previously described alterations of JAK1/JAK3/STAT5.

## Figures and Tables

**Figure 1 cancers-15-02527-f001:**
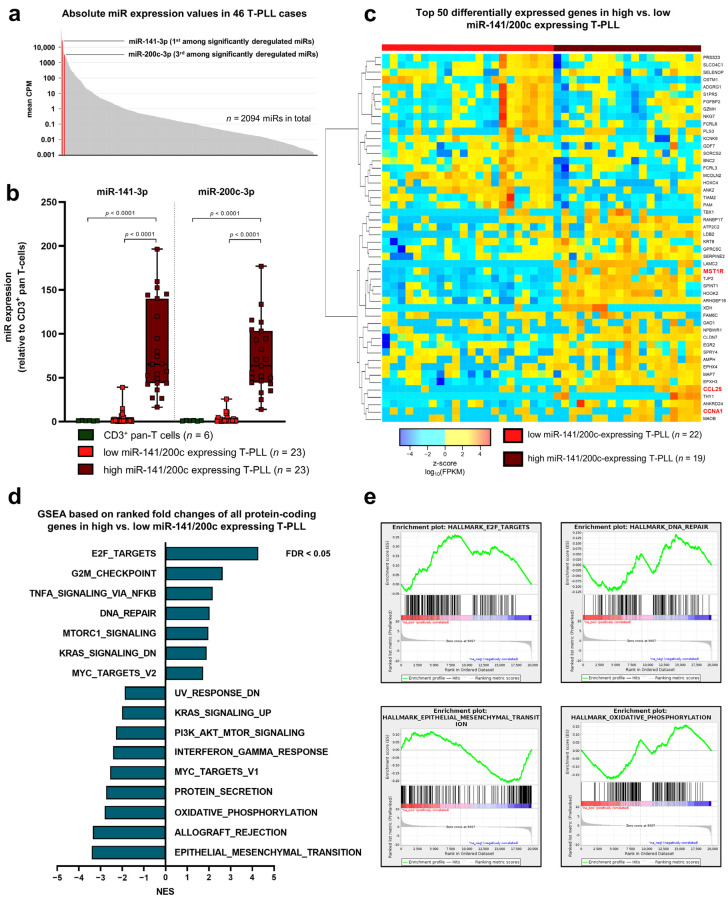
Upregulation of the miR-141/200c cluster, which is regularly seen in T-PLL cells, is associated with transcriptomes of accelerated cell cycle transition, enhanced survival signaling, and altered DNA damage responses in primary T-PLL. MicroRNA (miR) profiles were evaluated using small RNA sequencing of primary human peripheral blood-derived T-prolymphocytic leukemia (T-PLL) cells (*n* = 46 cases) and CD3^+^ healthy donor-derived pan-T-cell controls (*n* = 6 donors), recapitulating a previously published dataset [19]. In addition, polyA-RNA sequencing was performed on PB-isolated tumor cells from 41 miR-characterized T-PLL patients in order to align miR-141/200c expression data with those of global transcriptome alterations. (**a**) Bar chart displaying mean CPM values for a total of 2094 miRs detected by small RNA sequencing in at least one T-PLL patient. MiR-141-3p (mean CPM = 26,561) and miR-200c-3p (mean CPM = 3144) were among the miRs with the highest absolute CPM values and are highlighted in red. In total, 34 miRs were significantly deregulated in T-PLL cases compared to healthy donor-derived pan-T cells. (**b**) Relative expression of miR-141-3p and miR-200c-3p as analyzed by small RNA sequencing (RNA-seq) in primary T-PLL cells (*n* = 46) and healthy donor-derived CD3^+^ pan-T cells (dark green, *n* = 6). MiR-141-3p (fold change (fc) = 43.2; false discovery rate (FDR) = 0.005) and miR-200c (fc = 38.2; FDR = 0.005) are highly overexpressed in T-PLL cells. Notably, half of all sequenced T-PLL cases displayed miR-141/200c expression similar to healthy donor-derived T-cells (‘low miR-141/200c’ T-PLL subset, light red, *n* = 23 cases; for miR-141-3p: fc = 4.37; *p* = 0.98; for miR-200c-3p: fc = 3.81, *p* = 0.97; both one-way ANOVA, compared to healthy controls), and the other half showed marked upregulation of both miR-141-3p and miR-200c-3p (‘high miR-141/200c’ T-PLL subset, dark red, *n* = 23 cases; for miR-141-3p: fc = 82.08; *p* < 0.0001; for miR-200c-3p: fc = 72.58, *p* < 0.0001; both one-way ANOVA, compared to healthy controls). (**c**) Heatmap of the 50 highest differentially expressed mRNAs, comparing the ‘high miR-141/200c’ T-PLL subset (*n* = 22) to the ‘low miR-141/200c’ subset (*n* = 19). The colors represent z-scores of respective Fragments Per Kilobase Million (FPKM) values calculated for each mRNA (red = higher z-score; blue = lower z-score). (**d**,**e**) Gene set enrichment analysis (GSEA), based on the complete list of fold changes of all protein-coding genes comparing the ‘high miR-141/200c’ T-PLL subset (*n* = 22) to the ‘low miR-141/200c’ subset (*n* = 19). In total, 14 of 50 *HALLMARK* gene sets displayed significant positive enrichment (FDR < 0.05, Kolmogorov–Smirnov test) in high miR-141/200c-expressing T-PLL cases compared to their low miR-141/200c counterpart, whereas 12 gene sets showed significant negative enrichment in the high miR-141/200c cluster. A selection of *HALLMARK* pathways, related to cancer, is displayed in (**d**), and exemplary GSEA plots are presented in (**e**). Appendix A presents patient characteristics of all included T-PLL cases (*n* = 103). Appendix A summarizes differential gene expression and deregulated GSEA pathways between high and low miR-141/200c T-PLL cases.

**Figure 2 cancers-15-02527-f002:**
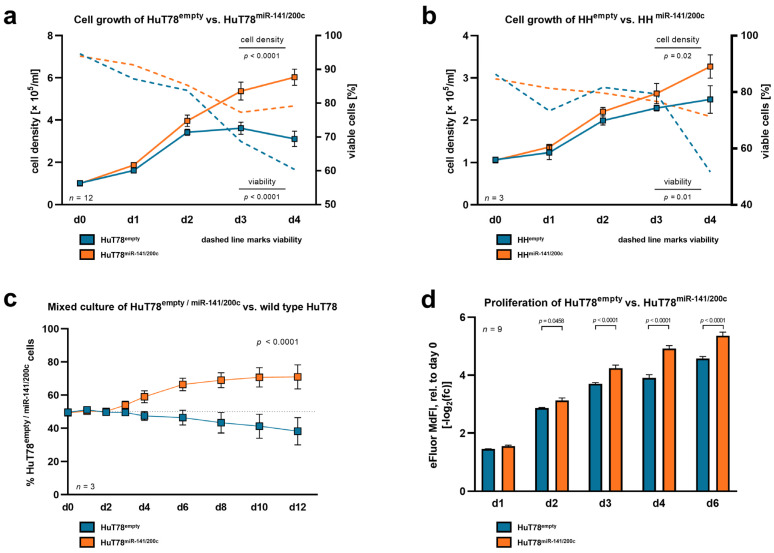
Overexpression of the miR-141/200c cluster in mature T-cell lymphoma lines leads to accelerated proliferation and diminished cell death under serum starvation. To interrogate whether higher levels of miR-141/200c lead to a more aggressive and proliferative cellular phenotype, we introduced lentiviral vectors with or without the miR-141/200c genomic cluster into the mature CTCL lines HuT78 and HH. (**a**,**b**) Cell density and viability as measured by Trypan blue staining of (**a**) HuT78^miR-141/200c^ and HuT78^empty^, and (**b**) HH^miR-141/200c^ and HH^empty^. Cells with and without miR-141/200c overexpression were seeded at a low density (1 × 10^5^ cells/mL) and cultured over four days in a culture medium with low serum contents (RPMI + 1% FBS (HuT78) or RPMI + 2% FBS (HH)). On each day, means with standard error of the mean (SEM) are presented. HuT78 and HH cells with high miR-141/200c expression partially overcame the stress induction via serum starvation, whereas the control cells showed stagnation in cell counts and more rapidly decreasing cell viability (HuT78: viability: *p* < 0.0001, cell density: *p* < 0.0001; HH: viability: *p* = 0.01; cell density: *p* = 0.02; two-way ANOVA). (**c**) Representative mixed culture of *n* = 3 independent experiments. HuT78^miR-141/200c^ or HuT78^empty^ cells (GFP-positive cells) were cultured in a 50:50 ratio together with parental, non-transduced HuT78 cells (GFP-negative cells). Cells were seeded in triplicate at a density of 3 × 10^5^/mL in growth medium with low serum contents (RPMI + 1% FBS). Culture medium with 1% FBS was replaced every two days. The GFP ratio was monitored over 12 days via flow cytometry. The light grey line marks the 50:50 ratio. HuT78^miR-141/200c^ outgrew their parental counterpart (d12: 69.88% GFP positivity), while the HuT78^empty^ control cells did not (d12: 40.18% GFP positivity; *p* < 0.0001, two-way ANOVA). (**d**) Assessment of cell proliferation in HuT78^miR-141/200c^ and HuT78^empty^ cells. HuT78^miR-141/200c^ and HuT78^empty^ cells were labeled with 5 μM of Invitrogen Cell Proliferation Dye eFluor 670^TM^ (ThermoFisher, Waltham, MA, USA), according to the manufacturer’s instructions. Then, cells were seeded at a low density (1 × 10^5^ cells/mL) and cultured over six days in culture medium with low serum contents (RPMI + 1% FBS). Signal intensity was assessed via flow cytometry. The signal intensity diminished over time, as labeled cells disperse their fluorescent dye to daughter cells with each cell division. The median fluorescence intensity (MdFI) for each time point was normalized to its own initial MdFI and -log2-transformed for data analysis and visualization. Mean values are presented for each day (mean with SEM, two-way ANOVA). In line, labeled HuT78^miR-141/200c^ cells presented lower MdFIs after two days as compared to HuT78^empty^ cells. The upregulation of the miR-141/200c cluster in the respective cell lines and a comparison of their miR-141/200c expression to that of primary T-PLL and healthy donor-derived T cells are presented in Appendix A. Functional assessments upon miR-141/200c upregulation in the naïve, T-acute lymphocytic leukemia (T-ALL)-like cell lines are presented in Appendix A (MOLT-4) and S1d (SUP-T11).

**Figure 3 cancers-15-02527-f003:**
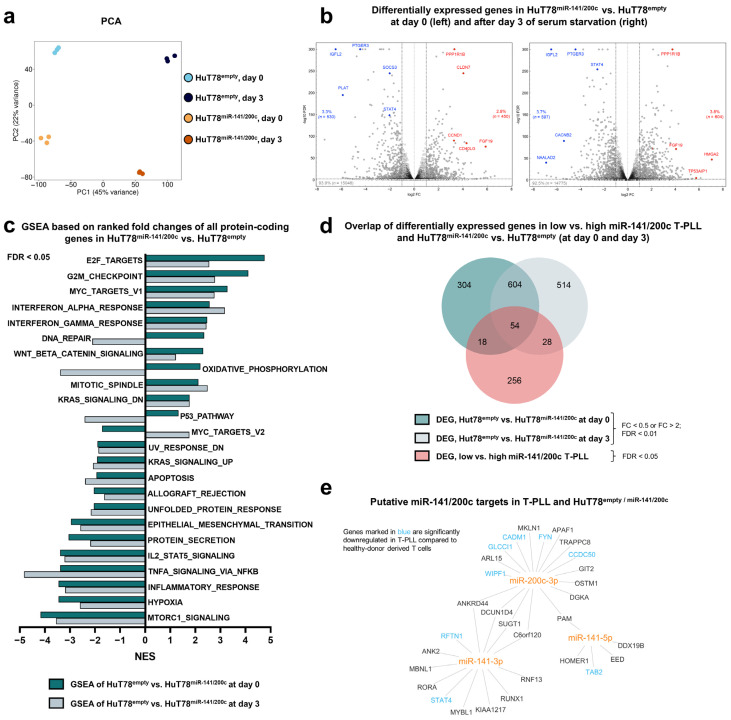
Overexpression of miR-141/200c shapes a pro-oncogenic transcriptome of the mature T-cell lymphoma line HuT78, resembling alterations seen in T-PLL cases. Analyses of differentially expressed genes comparing HuT78^miR-141/200c^ vs. HuT78^empty^ cells, without serum starvation (day 0 (d0)) and after three days of serum starvation (RPMI + 1% FBS; d3). For each time point, three technical replicates were sequenced for each transduced cell line. (**a**) Principal component analysis (PCA) based on all mRNAs. Separate clustering of the four different conditions indicated global differences in the transcriptome, which were induced by upregulation of the miR-141/200c cluster (principal component 2, 22% variance) and by serum starvation (principal component 1, 45% variance). (**b**) Volcano plots of all expressed mRNAs comparing HuT78^miR-141/200c^ vs. HuT78^empty^ cells at d0 (left) and d3 (right). The horizontal dashed line indicates an FDR of 0.01. The black vertical dashed lines mark an fc of 0.5 and 2; the light grey vertical line marks an fc of 1. Exemplary genes are highlighted in blue (downregulation) or red (upregulation) [31,32]. (**c**) GSEA (*HALLMARK* gene sets), based on the complete list of fold changes of all protein-coding genes comparing HuT78^miR-141/200c^ vs. HuT78^empty^ cells (FDR < 0.05, Kolmogorov–Smirnov test). The dark green bars indicate differentially enriched *HALLMARK* gene sets at d0, and the light green bars indicate them at d3. (**d**) Venn diagram showing overlapping genes comparing HuT78^miR-141/200c^ vs. HuT78^empty^ cells at d0 and d3 (fc < 0.5 or fc > 2; FDR < 0.01), as well as the high miR-141/200c T-PLL subset vs. the low-miR141/200c T-PLL subset (FDR < 0.05). In total, 54 genes presented a differential expression in all three conditions. (**e**) Network of predicted targets (by seed sequences, see Section 2 for details), correlating negatively with miR-141/200c expression in miR-141/200c-modulated HuT78 cells and in T-PLL cells (*p* < 0.05, Spearman correlation). Font color represents a differential expression of miR/mRNA comparing T-PLL cells (*n* = 48 cases) and healthy donor-derived CD3^+^ pan-T cells (*n* = 6 donors; blue = lower expression; red = higher expression, based on our previously published RNA-seq data [6]). *STAT4* emerges as a target of the miR-141/200c cluster. Heatmaps showing the top 50 differentially expressed genes comparing HuT78^miR-141/200c^ vs. HuT78^empty^ cells at d0 and d3, as well as heatmaps of core-enriched genes of exemplary *HALLMARK* gene sets are presented in Appendix A. Differentially expressed genes and respective GSEAs upon serum starvation are displayed in Appendix A. Appendix A summarizes differential gene expression and deregulated GSEA pathways between HuT78^empty^ and HuT78^miR-141/200c^ cells at day 0 and day 3. Appendix A summarizes differential gene expression and deregulated GSEA pathways at day 3 vs. at day 0 of serum starvation in HuT78^empty^ or HuT78^miR-141/200c^ cells.

**Figure 4 cancers-15-02527-f004:**
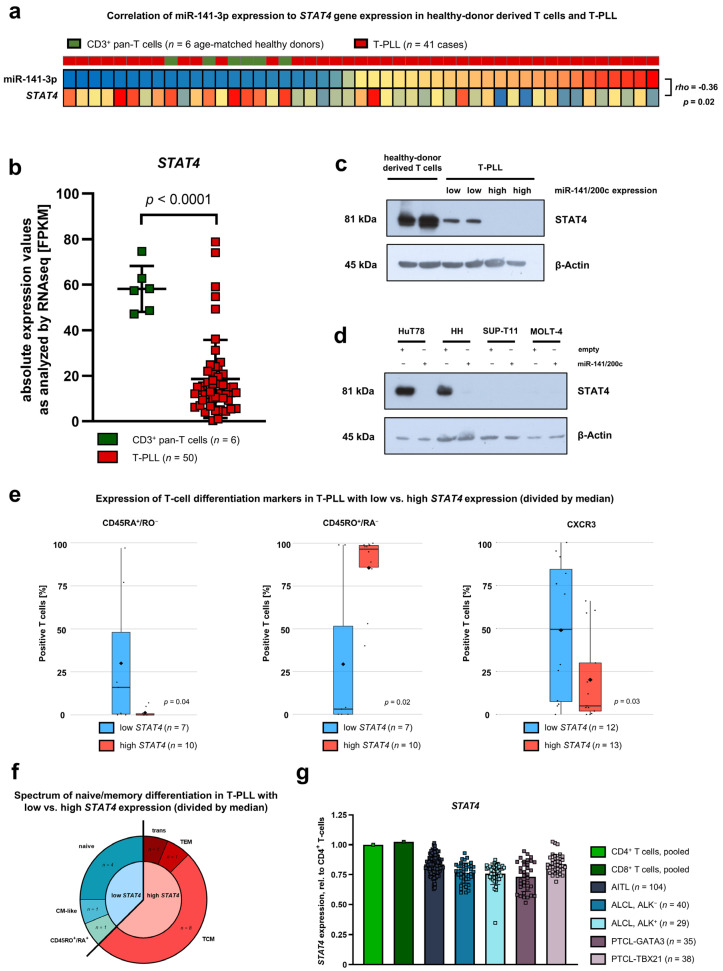
Downregulation of STAT4 by the miR-141/200c cluster is associated with an immature phenotype in T-PLL cells. (**a**) Heatmap presenting *STAT4* mRNA expression and miR-141-3p expression for T-PLL cases (*n* = 41) and pan CD3^+^ T cells of age-matched healthy donors (*n* = 6). Color codes represent z-scores based on means of the respective data dimension. *STAT4* mRNA expression showed a significant, inverse correlation with miR-141-3p expression (Spearman correlation coefficient = −0.36, *p* = 0.02). (**b**) Absolute expression values of *STAT4* mRNA as analyzed by RNA-seq in primary T-PLL cells (*n* = 50) and age-matched, healthy donor-derived CD3^+^ pan-T cells (*n* = 6). FPKM values are displayed. The RNA-seq dataset was previously published by us [19]. The potential miR-141/200c target *STAT4* is significantly downregulated in T-PLL cases (*p* < 0.0001, unpaired t-test). (**c**) Immunoblot presenting STAT4 protein expression in pan CD3^+^ T cells of two age-matched healthy donors, two low miR-141/200c-expressing T-PLL cases, and two high miR-141/200c-expressing T-PLL cases. β-Actin was used as the housekeeping protein. We identified globally diminished STAT4 protein levels in T-PLL cells when compared to healthy donor-derived T cells. (**d**) Immunoblot presenting STAT4 protein expression in HuT78, HH, MOLT-4, and SUP-T11 cells, in each case in the empty-vector-transduced cell lines as well as in the miR-141/200c-overexpressing condition. β-Actin was used as the housekeeping protein. (**e**) Association analyses of *STAT4* mRNA expression with surface markers of T-cell differentiation in primary T-PLL cells. Groups of low and high *STAT4* expression were assigned by results of GEP array analyses [6] and divided by the median. After division into two groups, cases with lower (blue) were compared to those with higher (red) *STAT4* expression. Cases were evaluated for CD45RO, CD45RA, and CXCR3 surface expression using flow cytometry. The median expression of the respective parameter is shown by a horizontal, back line; the mean expression by a diamond symbol (♦).(**f**) Association analyses of *STAT4* mRNA expression with the spectrum of naïve/memory differentiation, based on the expression of CD45RA, CD45RO, CCR7, and CD62L, in primary T-PLL cells. Groups of low and high *STAT4* expression were assigned by results of GEP array analyses [6] and divided by the median. After division into two groups, cases with lower (blue) were compared to those with higher (red) *STAT4* expression. A 70% cutoff was used to classify the predominant differentiation. Naïve T-cells were defined as CD45RA^+^/CD45RO^−^ T-cells, central memory T-cells (TCM) as CD45RO^+^/CCR7^+^/CD62L^+^ T-cells, effector memory T-cells (TEM) as CD45RO^+^/CCR7^−^/CD62L^−^ T-cells, transitional T-cells (trans) as CD45RO^+^/CCR7^+^/CD62L^−^ T-cells, and central-memory-like (CM-like) T-cells as CD45RO^−^/CD45RA^−^/CCR7^+^/CD62L^+^. (**g**) Relative expression of *STAT4* mRNA as analyzed by GEP arrays in angioimmunoblastic T-cell lymphoma (AITL, *n* = 104), ALK^+^ (*n* = 29) and ALK^-^ anaplastic large-cell lymphoma (ALCL, *n* = 40), as well as in peripheral T-cell lymphoma (PTCL)-GATA3 (*n* = 35) and PTCL-TBX21 (*n* = 38), utilizing previously published datasets [25,26]. Pooled samples of CD4^+^ and CD8^+^ T-cells of healthy donors were used as controls. *STAT4* mRNA expression in high vs. low miR-141/200c T-PLL, as well as associations of *STAT4* mRNA expression with CXCR5 and GATA3 surface expression in T-PLL, are presented in Appendix A. Complete scans of the immunoblots shown in Figure 4c,d are presented in Appendix A.

**Figure 5 cancers-15-02527-f005:**
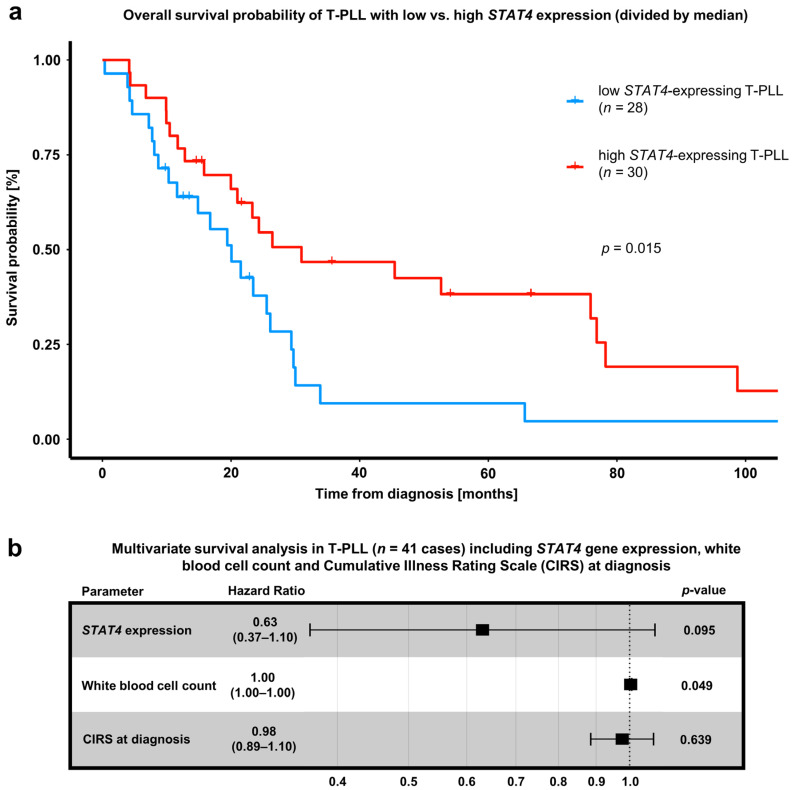
Downregulation of *STAT4* mRNA is associated with poor patient outcomes in T-PLL. (**a**) Overall survival (OS), comparing T-PLL cases with *STAT4* mRNA expression higher than the median *STAT4* expression of all T-PLL cases (*n* = 28) to those with lower *STAT4* mRNA expression than the median (*n* = 30). For the evaluation of *STAT4* mRNA expression, our previously published gene-expression array data were utilized [6]. Notably, the OS of low *STAT4* mRNA-expressing cases (median OS: 20.1 months) was shorter compared to cases with high *STAT4* mRNA expression (median OS: 31.0 months, *p* = 0.02, log-rank test. (**b**) Multivariate survival analysis in 41 T-PLL cases, assessing the impact of *STAT4* gene expression (by gene-expression array data [6]), white blood cell (WBC) count, and Cumulative Illness Rating Scale (CIRS) at diagnosis. Statistical significance was assessed via Cox regression analysis. Outcomes after alemtuzumab-based therapy in association with miR-141/200c or *STAT4* expression are presented in Appendix A.

## Data Availability

Sequencing data on primary T-PLL samples and primary TCL are published elsewhere [19,25,26]. Sequencing data on HuT78 cells are available on request from the corresponding author.

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
