# Peer review of "The miR-141/200c-STAT4 Axis Contributes to Leukemogenesis by Enhancing Cell Proliferation in T-PLL"

_cancers, 2023, doi:10.3390/cancers15092527_

Round 1
Reviewer 1 Report
The manuscript by Otte and colleagues describes an interesting oncogenic mechanism of the miR-141/200c cluster in T-prolymphocytic leukemia (T-PLL). Through comprehensive analyses of dysregulated miRs and mRNAs in T-PLL patient samples as well as cellular models of mature T-cell cancer, the authors report the upregulation of miR-141/200c in T-PLL and demonstrate that increased miR-141/200c expression confers a cell proliferative/survival advantage and correlates with reduced STAT4 expression and a more immature T-cell differentiation state. Notably, this work defines STAT4 expression as predictive of T-PLL patient survival, with important prognostic value. The authors state that more detailed analyses of the direct regulation of STAT4 by miR-141/200c or other factors will be the topic of future studies.
This manuscript is very well written and presented, and would be acceptable for publication in its current form.
Minor comments:
- it would be helpful to the reader to state at the beginning of section 3.2 that, because currently no human T-PLL cell lines exist, the CTCL and T-ALL cell lines were selected for further studies.
- information on the culture conditions (media supplements, cytokines, etc) could not be found for the primary T-PLL and control CD3+ T-cells in the supplementary information (or in the main methods or figure legends). Details pertaining to this should be included.
- to refine the extensive figure legends, it should be noted that figure legends normally should not contain statements or interpretation of the results. Only key methodological, experimental and statistical information should be given, and interpretation of the results can be left to the results text.
Reviewer 2 Report
T-prolymphocytic leukemia(T-PLL) is a relatively rare and poor prognosis mature T-cell malignancy disease. In this study, the authors identified an increasing expression of miR-141/200c cluster in T-PLL cells, which other group has demonstrated the overexpression of miR-200c/141 correlated with TGFβ pathway affected in T-PLL(https://doi.org/10.3324/haematol.2020.263756). The authors indicated that miR-141/200c regulate several genes expression involved in cell survival and differentiation. And they identified STAT4 as a potential miR-141/200c target gene, which associated with an immature phenotype of primary T-PLL cells as well as with a shortened overall survival of T-PLL patients.
Although the paper is well organized and the topic is interesting, some comments need to be addressed to improve the overall study design:
Major:
1. The author indicates that miR-141/200c could regulate the expression of STAT4, but it is not clear whether this regulation is direct or indirect. If the author’s hypothesis is miR-141/200c directly down-regulate STAT4, it is necessary to verify the binding sites of miR-141/200c in the 3'UTR region of STAT4 and to confirm it by using the Luciferase Reporter system with mutants of specific sites.
2. The authors focused on STAT4 expression regulated by miR-141/200c significantly. Please explain why STAT4 is not included in the top 50 genes in Fig. 1c?
3. More evidence are needed to address that downregulation of STAT4 by miR-141/200c cluster is associated with an immature cell phenotype and poor patient outcomes in T-PLL (Fig 4). For example, how is the role of STAT4 on cell proliferation of T-PLL cells? Does overexpression of miR-141/200c resistance STAT4 could rescue the phenotype of accelerated cell proliferation rate in overexpression of miR-141/200c T-PLL cell lines?
Minor:
1. miR-141/200c cluster is associated with poor patient outcomes in T-PLL. Did the author test if knocking down miR-141/200c could affect the cell proliferation in T-PLL cell lines? Does the author double check with the basal level of miR-141/200c in the cell lines as well as the primary T-PLL cells? Does the expression of miR-141/200c negatively corelated with the expression of STAT4 in these four cell lines?
2. Currently, the best treatment for T-PLL is intravenous alemtuzumab, which results in high response rates and a significant improvement in survival. Could the authors investigate if miR-141/200c cluster involved in the response to this therapy?
